# Recurrent rearrangements of *FOS* and *FOSB* define osteoblastoma

Matthew W. Fittall [1,2,3], William Mifsud[2,3,4], Nischalan Pillay [2,5], Hongtao Ye[5], Anna-Christina Strobl[5], Annelien Verfaillie[1], Jonas Demeulemeester [1,6], Lei Zhang[7], Fitim Berisha[5], Maxime Tarabichi[1,3], Matthew D. Young[3], Elena Miranda[2], Patrick S. Tarpey[3], Roberto Tirabosco[5], Fernanda Amary[5], Agamemnon E. Grigoriadis[8], Michael R. Stratton[3], Peter Van Loo [1,6], Cristina R. Antonescu[7], Peter J. Campbell[3], Adrienne M. Flanagan[2,5] & Sam Behjati[3,9]

The transcription factor *FOS* has long been implicated in the pathogenesis of bone tumours, following the discovery that the viral homologue, *v-fos*, caused osteosarcoma in laboratory mice. However, mutations of *FOS* have not been found in human bone-forming tumours. Here, we report recurrent rearrangement of *FOS* and its paralogue, *FOSB*, in the most common benign tumours of bone, osteoblastoma and osteoid osteoma. Combining whole-genome DNA and RNA sequences, we find rearrangement of *FOS* in five tumours and of *FOSB* in one tumour. Extending our findings into a cohort of 55 cases, using FISH and immunohistochemistry, provide evidence of ubiquitous mutation of *FOS* or *FOSB* in osteoblastoma and osteoid osteoma. Overall, our findings reveal a human bone tumour defined by mutations of *FOS* and *FOSB*.

[1] The Francis Crick Institute, London NW1 1AT, UK. [2] University College London Cancer Institute, London WC1E 6DD, UK. [3] Wellcome Trust Sanger Institute, Hinxton, Cambridgeshire CB10 1SA, UK. [4] Great Ormond Street Hospital for Children NHS Foundation Trust, London WC1N 3JH, UK. [5] Department of Histopathology, Royal National Orthopaedic Hospital NHS Trust, Stanmore, Middlesex HA7 4LP, UK. [6] Department of Human Genetics, University of Leuven, Leuven 3000, Belgium. [7] Department of Pathology, Memorial Sloan Kettering Cancer Center, New York, NY 10065, USA. [8] Centre for Craniofacial and Regenerative Biology, King's College London, Guy's Hospital, London SE1 9RT, UK. [9] Department of Paediatrics, University of Cambridge, Cambridge CB2 0QQ, UK. These authors contributed equally: Matthew W. Fittall, William Mifsud. These authors jointly supervised this work: Adrienne M. Flanagan, Sam Behjati. Correspondence and requests for materials should be addressed to A.M.F. (email: a.flanagan@ucl.ac.uk) or to S.B. (email: sb31@sanger.ac.uk)

Osteoblastoma is the most common benign bone-forming tumour. It typically occurs in the medulla of long bones and the neural arch from where it may extend into the vertebral body[1]. Osteoid osteoma is thought to represent a variant of osteoblastoma. The two entities are distinguished arbitrarily by size, with osteoblastoma measuring more than 2 cm in diameter. Large, inaccessible and recurrent tumours can cause considerable morbidity[1]. Treatment is by surgical resection. The genetic changes underpinning osteoblastoma have been studied at the resolution of karyotypes and copy number arrays. Copy number losses involving chromosome 22 and rearrangements involving chromosome 14 have been reported in rare cases only[2,3].

Diagnosis of osteoblastoma is currently based on histological assessment. Occasionally this can be challenging, as osteoblastoma has to be distinguished from osteoblastic osteosarcoma, an aggressive bone cancer that requires extensive, sometimes disabling, multimodal treatment[4].

Here, we sought to define the somatic changes that underpin osteoblastoma. Our starting point was a discovery cohort of six tumours, five osteoblastomas and one osteoid osteoma, that we subjected to RNA and whole-genome DNA sequencing. Tissue was obtained from frozen tumour and corresponding germline DNA sequences derived from blood samples. Using the analysis pipeline of the Cancer Genome Project ('Methods'), we generated catalogues of all classes of somatic mutations: substitutions, indels, structural variants (rearrangements) and copy number changes. Transcriptome sequences were analysed to corroborate DNA changes and to call gene fusions.

Our key finding was recurrent, disease-defining structural variation of the *FOS* and *FOSB* oncogenes in osteoblastoma and osteoid osteoma.

## Results

**Osteoblastoma habours few somatic alterations**. Overall, there was a paucity of somatic alterations in osteoblastoma, with a median mutation burden of 319 substitutions per genome (range, 123–700) and 28 indels per genome (range, 14–50; Supplementary Data 1–3). Similarly, copy number analyses demonstrated diploid tumours with few aberrations (Supplementary Fig. 1 and Supplementary Data 4). The previously reported losses in chromosome 22 were not seen in our cases[2]. Only a small number of mutations affected the coding sequence of genes, none of which were plausible driver events.

**Recurrent *FOS* and *FOSB* rearrangements**. Against this backdrop of a quiet somatic architecture, analysis of structural variants revealed break points in the AP-1 transcription factor *FOS*, in 5/6 cases, and its paralogue *FOSB* in the sixth case (Figs. 1 and 2; and Supplementary Data 5). We analysed and validated these rearrangements at the DNA level by local assembly, copy number analyses and at the RNA level by identification of break point spanning cDNA reads (Supplementary Data 6–8). A single *FOS* or *FOSB* break point was confirmed in each sample, suggesting that these were mono-allelic rearrangements. There was no evidence of similar rearrangements in paired normal tissue samples, confirming that they were somatic. *FOS* rearrangements were also validated with Sanger sequencing of cDNA (Supplementary Fig. 2).

Unusually for structural variants, all *FOS* break points were exonic, residing within a narrow genomic window of exon 4 (Fig. 1a). The rearrangements comprised both interchromosomal and intrachromosomal events. The rearrangement partners were introns of other genes (3/5 cases) or intergenic regions (2/5 cases). There was evidence of expression of the fusion transcript, visible as aberrant spikes in RNA-Seq read coverage adjacent to the break point in the fusion partners. However, these aberrantly

transcribed sequences did not include any known exonic sequence. Indeed, stop codons were encountered at, or immediately after the break points (Fig. 1d and Supplementary Fig. 2).

*FOSB* rearrangements have been described in two different types of vascular tumours, namely pseudomyogenic haemangioendothelioma and epithelioid haemangioma[5,6]. The interchromosomal translocation, found in PD7525a, occurred in the same region of exon 1 (Fig. 2). cDNA reads spanning the fusion junction support the expression of a fusion gene, connecting, in frame, *PPP1R10* to *FOSB*. Consequently, the expression of the *FOSB* fusion gene would be brought under the control of the *PPP1R10* promoter (Supplementary Fig. 3).

In contrast to the *FOSB* genomic alteration, the rearrangements of *FOS* do not involve coding sequence of other genes. Transcription remained under the control of its native promoter. Furthermore, in 2/5 cases the fusion partner did not lie within a gene. These observations are supported by re-analyses of RNA sequences of epithelioid haemangioma harbouring *FOS* rearrangements (Fig. 1d)[7,8]. Similarly to osteoblastoma, the break points in these vascular tumours clustered within the same narrow 200 bp window of exon 4. Furthermore, stop codons were again found in the immediate vicinity of the *FOS* break point.

**FOS and FOSB alterations are ubiquitous in osteoblastoma**. To validate our findings, we examined by fluorescence in situ hybridisation (FISH) an extension cohort of 55 formalin-fixed paraffin-embedded (FFPE) histologically typical cases of osteoblastoma and osteoid osteoma (Supplementary Data 1). In these 55 samples, we found *FOSB* and *FOS* breakapart signals in 1 and 48 tumours, respectively (89% in total; Supplementary Data 1).

We speculated that the six FISH-negative cases may also harbour *FOS* or *FOSB* rearrangements that were not detected because FISH analysis is hampered in tumours of low cellularity, a frequent feature of osteoblastoma[1]. FISH may also miss cases with short distance intrachromosomal rearrangements, such as tandem duplications, that insufficiently separate probe target sequences. Since sufficient tissue was available for 3/6 negative cases, we sought alternative evidence for *FOS* dysregulation by immunohistochemistry. All three samples demonstrated strong nuclear FOS immunoreactivity, supporting the notion that alterations in *FOS* or *FOSB* underpin every case of osteoblastoma and osteoid osteoma (Supplementary Fig. 4b). FOSB immunohistochemistry was uninformative in osteoblastoma, consistent with previous experience with decalcified tumours (Supplementary Fig. 4c)[9].

**FOS and FOSB alterations are specific to benign bone tumours**. To explore the utility of our findings as diagnostic markers of osteoblastoma, we assessed their specificity across different tumour sets. We examined FOS immunoreactivity in 183 cases of osteosarcoma, including 97 cases of osteoblastic osteosarcoma, and 17 cases of angiosarcoma. In keeping with previous reports, FOS immunoreactivity was seen in osteosarcoma samples but only one had a distribution and intensity of immunoreactivity comparable with osteoblastoma[10]. While there were no breakapart signals in *FOS* or *FOSB* on FISH testing, copy number gains were noted (Supplementary Fig. 4d). We then examined 55 whole-genome sequences of two published osteosarcoma series, none of which harboured break points in *FOS* or *FOSB*[11,12]. Finally, we could not find similar *FOS* and *FOSB* rearrangements in whole-genome sequences in 2652 non-osteoblastoma tumours[13]. Taken together, our findings indicate that *FOS* and *FOSB* alterations may be exploited as diagnostic markers for osteoblastoma and osteoid osteoma. We also demonstrate for the first time that both tumour types are similar at a molecular level.

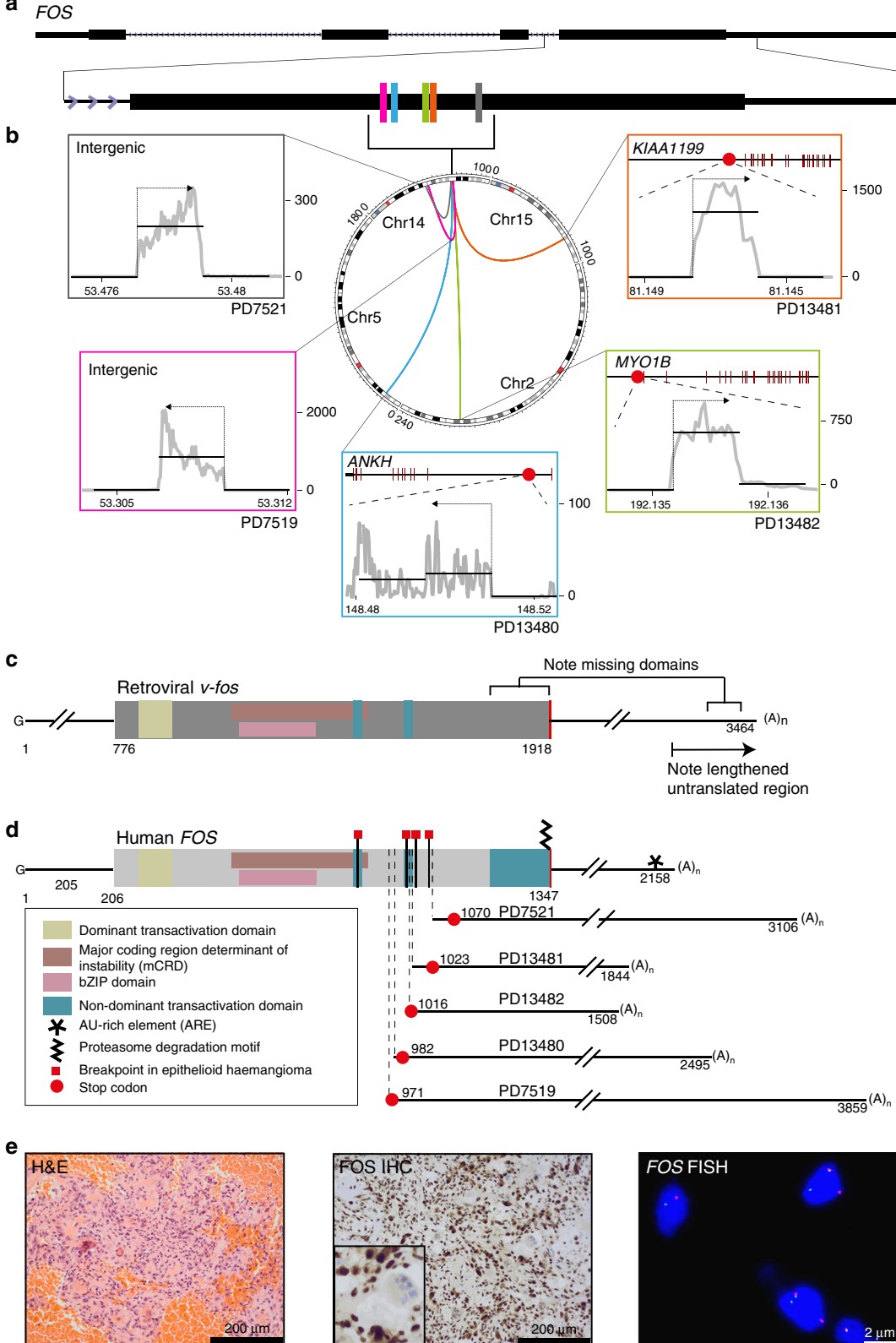

**Fig. 1** *FOS* fusions in osteoblastoma. **a** Clustered break points in *FOS*. **b** Central Circos plot showing the clustering of break points in *FOS*-mutant samples. All structural variants involving chromosome 14 are shown, demonstrating the paucity of genomic rearrangements. Surrounding panels demonstrate normalised RNA-Seq read counts for each fusion partner. Horizontal line segments reflect mean sequencing counts. The arrow above shows the direction of transcription of the fusion. **c** Retroviral *v-fos*. **d** Schematic of *FOS* break points in benign bone and vascular tumours generating similarity with the murine retroviral transforming *v-fos*. Subscript numbers from left to right report the length of the transcript to the stop codon and the predicted cleavage and poly-adenylation site, respectively. **e** FOS immunohistochemistry demonstrating strong nuclear immunoreactivity in *FOS*-mutant osteoblastoma, PD13482 (centre with zoom inset), Haematoxylin and eosin (H&E) (left), and a clear breakapart of FISH probes surrounding the *FOS* locus (right)

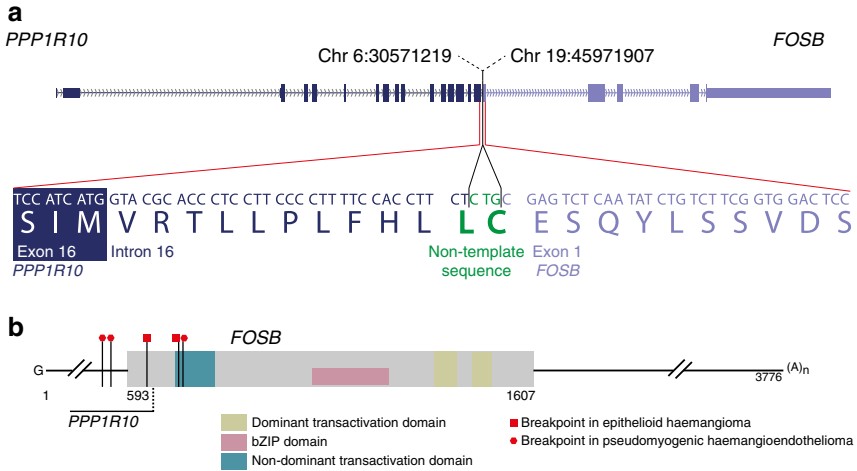

**Fig. 2** *FOSB* fusion in PD7525. **a** Schematic of the *PPP1R10-FOSB* fusion. The *PPP1R10* exon 16 splice donor site is skipped, resulting in exonisation of part of the intron. A three-nucleotide non-template sequence at the break point then allows an in-frame *PPP1R10-FOSB* fusion transcript to be produced. **b** Annotated schematic of the wild-type *FOSB* transcript with the published break points in the vascular tumours (pseudomyogenic haemangioendothelioma (PHE), epithelioid haemangioma (EH))

## Discussion

Rearrangements of *FOS* moulded a mutant transcript that lacks regulatory elements. This configuration bears a striking resemblance to the retroviral oncogene, *v-fos*, identified in the FBJ murine osteosarcoma virus (Fig. 1c, d). Dysregulated expression of the murine orthologue, *c-fos*, can cause osteosarcoma in model systems but requires fusion with a highly active promoter and the *v-fos* 3′ untranslated region[14].

FOS levels are tightly regulated by both transcript and protein degradation. Two translation-dependent mechanisms ensure rapid mRNA degradation: a length-dependent interaction between the poly-A tail and an exon 3 domain (known as the major coding region determinant of instability)[15], and an independent AU-rich element in the 3′ untranslated region[16]. Both mechanisms are likely to be disrupted by the rearrangements we have found. Furthermore, ubiquitin-independent proteasomal degradation rapidly depletes the wild-type FOS protein[17]. The C-terminal truncations seen in epithelioid haemangioma have recently been shown to protect FOS from degradation[18]. While the break points disrupt components of the C-terminal transactivation domain, this is not required for in vitro transformation by *v-fos*[19,20]. While we cannot also exclude alteration of AP-1 activity we would expect increased FOS concentration in osteoblastoma cells. Consistent with this prediction, we observed intense nuclear immunoreactivity of FOS in osteoblastoma cells (Fig. 1f and Supplementary Data 1). Our findings may explain the absence of nonsense mutations in *FOS*, as only rearrangements could abolish both levels of regulation.

Fifty years after the identification of *v-fos* we report human bone-forming tumours, osteoblastoma and osteoid osteoma, that are predominantly characterised by an aberrant *FOS* homologue resembling the viral *fos* oncogene. This shifts our understanding of *FOS*/AP-1 dysregulation in human bone tumours. Our findings also draw an intriguing parallel between bone-forming tumours and a subset of vascular tumours, suggesting possible shared developmental pathways. Patients are likely to benefit from our findings, as they can be readily translated into routine diagnostic practice.

## Methods

**Patient samples**. Patients provided their written and informed consent to provide samples for this study, which was approved by the National Research Ethics Service (NRES) Committee Yorkshire and The Humber – Leeds East (15/YH/0311).

**Sequencing**. Tumour DNA and RNA were derived from fresh-frozen tissue reviewed by bone pathologists (A.M.F./R.T./F.A.). Matched normal DNA was acquired from blood samples. Whole-genome sequencing was performed using the Illumina HiSeq 2000 or 2500 platform, using 100 bp paired-end sequencing. For whole-genome sequencing, we followed the Illumina no-PCR library protocol to construct short insert 500 bp libraries, prepare flowcells and generate clusters. The average coverage of tumours was at least 40× and of normal DNA at least 30× after alignment with BWA-Mem (2.0.54)[21] (Supplementary Data 9). Poly-A RNA was sequenced on an Illumina HiSeq 2000 (75 bp paired-end). Sequenced RNA libraries were aligned with STAR (2.0.42)[22].

**Variant detection**. The Cancer Genome Project (Wellcome Trust Sanger Institute) variant calling pipeline was used to call somatic mutations. The following algorithms, with standard settings, and no additional post-processing was used on aligned DNA BAM files: CaVEMan (1.11.0)[23] for substitutions; Pindel (2.1.0)[24] for indels; BRASS (5.3.3 https://github.com/cancerit/BRASS) for rearrangements, and ASCAT NGS (4.0.0)[25] for copy number aberrations. Aligned RNA BAM files, including those realigned from published data, were run through the RNA-Seq analysis pipeline (https://github.com/cancerit/cgpRna/wiki), which includes HTSeq (0.6.1)[26] for gene feature counts, and the combination of STAR (2.5.0c), TopHat2 (2.1.0)[27] and deFuse (0.7.0)[28] fusion discovery protocols.

**Variant validation**. The precision of Cancer Genome Project (Wellcome Trust Sanger Institute) variant calling pipeline has been determined in multiple studies[29]. We confirmed this through manual inspection of raw sequencing reads. As for rearrangements, we only included break points in this data set that had been validated by reconstruction at base pair resolution.

**Analysis of mutations in cancer genes**. We analysed variants using a previously documented strategy[12]. In brief, we considered variants as potential drivers if they presented in established cancer genes (COSMIC v82). Tumour suppressor coding variants were considered if they were annotated as functionally deleterious by the in-house algorithm, VAGrENT (http://cancerit.github.io/VAGrENT/). Disruptive rearrangement break points in or homozygous deletions of tumour suppressors were also considered. Additionally, homozygous deletions were required to be focal (<1 Mb in size). Mutations in oncogenes were considered driver events if they were located at previously reported hot spots (point mutations) or amplified the intact gene. Amplifications also had to be focal (<1 Mb) result in at least five copies in diploid genomes.

**Fusion detection**. Rearrangements in FOS and FOSB were analysed using the DNA structural rearrangement caller, BRASS and the in-house RNA fusion detection algorithm, infuse. Fusions were considered if break points and orientations were supported by both algorithms. All reads supporting the break points were manually inspected. In sample PD13482, in which neither algorithm identified the fusion, both split reads and discordant read pairs spanning the fusion were identified in the DNA- and RNA-Seq data.

All *FOS* fusion partner break points were located in genomic regions not normally represented in RNA sequencing libraries as they were intergenic or intronic segments. The per-base coverage in these regions therefore reveals a clear peak, present only in that tumour sample, demonstrating expression of aberrant

transcripts (normalised by the mean of HTSeq counts $\times 10^3$). The end of the transcript was considered to be 10–30 bp downstream of the cleavage and poly-adenylation signal ('AATAAA') with the greatest drop in coverage in the surrounding 200 bp. For schematic purposes, mean normalised coverage was plotted as a segment, as Fig. 1: the 'mate transcript segment' is between the break point (grey vertical dashed line) and the poly-adenylation cleavage site; surrounding segments are the mean sequencing coverage over a genomic range of equal length to the 'mate transcript segment'.

**FOS fusion validation**. cDNA was synthesised from 1 µg of total RNA was using the ProtoScript® II First-Strand cDNA Synthesis Kit (NEB). PCR was performed with Phusion high-fidelity PCR master mix (HF buffer, NEB) using the primers listed in Supplementary Data 10. Amplified products were size selected using gel electrophoresis and then Sanger sequenced using an internal primer listed in Supplementary Data 10.

**Allele-specific expression analysis**. We analysed allele-specific expression in *FOS* and *FOSB* using allele counts at heterozygous single-nucleotide polymorphisms (SNPs). To allow for poor alignment in RNA-Seq data close to break points, allele counts at heterozygous SNPs were computed manually. Heterozygous SNPs were identified from DNA sequencing data. Allele counts were measured from RNA-Seq reads using GATK ASEReadCounter[30].

**Fluorescence in situ hybridisation (FISH) for *FOS* and *FOSB***. A cohort of 55 informative cases of osteoblastoma/osteoid osteoma was examined by FISH for *FOS* breakapart. *FOSB* probes were custom designed with Agilent SureDesign to flank the breakapart region. *FOS* probes and methods have been described previously[8] (Supplementary Data 11). In brief, deparaffinised sections were pretreated by pressure cooking for 5 min and subsequently incubated in pepsin solution at 37 °C for 50 min. Probes were applied to tissue sections and denatured at 72 °C, followed by hybridisation overnight at 37 °C. After hybridisation, the sections were washed and mounted with 4′,6-diamidino-2-phenylindole and coverslips.

**Immunohistochemistry for FOS and FOSB**. Deparaffinised hydrated tissue sections underwent antigen unmasking in Tris-EDTA pH 9 (DAKO S2367 - Agilent Technologies LDA UK Limited, Cheshire, UK) at high pressure for 2 min. After washing and quenching, sections were blocked in 2.5% horse serum (Vector ImmPRESS Kit) for 20 min at room temperature. Incubation with primary antibodies was for 60 min, secondary antibodies for 30 min, and DAB + substrate/chromagen (Dako, K3468) for 5 min, all at room temperature, prior to counter-staining and mounting. FOS antibodies were EMD Millipore ABE457 (Rabbit Polyclonal, used at 1 or 0.5 µg mL$^{-1}$) and ImmPRESS Horse Radish Peroxidase Anti-Rabbit IgG (Peroxidase) Polymer Detection Kit, made in Horse (MP-7401, Vector Laboratories, Peterborough, UK) while FOSB antibodies (clone 5G4, dilution 1:100, Cell Signaling Technology, Danvers, MA and rabbit polyclonal CAMTA1 antibody Atlas Antibodies, Stockholm, Sweden) were as previously described[31].

**Data availability**. The authors declare that all data supporting the findings of this study are available within the article and its supplementary files or from the corresponding author on reasonable request. Sequencing data have been deposited at the European Genome-Phenome Archive (http://www.ebi.ac.uk/ega/) that is hosted by the European Bioinformatics Institute. DNA (https://www.ebi.ac.uk/ega/datasets/EGAD00001000785; https://www.ebi.ac.uk/ega/datasets/EGAD00001000147) accession numbers: EGAN00001100713, EGAN00001100730, EGAN00001100714, EGAN00001100731, EGAN00001100715, EGAN00001100732, EGAN00001031765, EGAN00001032117, EGAN00001031767, EGAN00001032119, EGAN00001036773, EGAN00001036983. RNA (https://www.ebi.ac.uk/ega/studies/EGAS00001000763) accession numbers: EGAN00001196539, EGAN00001196540, EGAN00001209957, EGAN00001196544, EGAN00001196545, EGAN00001209959.

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

## Acknowledgements

This work was funded by The Wellcome Trust; Skeletal Cancer Action Trust UK; the Royal National Orthopaedic Hospital NHS Trust; Rosetrees Trust. A.M.F. is an NIHR

senior investigator and was also supported by the National Institute for Health Research, UCLH Biomedical Research Centre and the UCL Experimental Cancer Centre. M.W.F., A.V., J.D., M.T. and P.V.L. are supported by the Francis Crick Institute, which receives its core funding from Cancer Research UK (FC001202), the UK Medical Research Council (FC001202) and the Wellcome Trust (FC001202). Personal fellowships have been granted to S.B. (Wellcome Trust Intermediate Clinical Research Fellowship; St. Baldrick's Foundation Robert J. Arceci International Innovation Award); P.J.C. (Wellcome Trust Senior Clinical Research Fellowship); N.P. (CRUK Clinician Scientist Fellowship) and E.M. (CRUK Career Development Fellow); C.R.A. is supported by NIH grants (P50 CA140146-01 and P30 CA008748), M.W.F. (Crick-CRUK Doctoral Fellowship); J.D. (Research Foundation – Flanders, FWO Postdoctoral Fellowship; European Union's Horizon 2020 Research and Innovation programme, MSCA 703594-DECODE); M.T. (European Union's Horizon 2020 Research and Innovation programme postdoctoral fellowship, MSCA 747852-SIOMICS). P.V.L. is a Winton Group Leader in recognition of the Winton Charitable Foundation's support towards the establishment of The Francis Crick Institute. We are grateful to Donna Magsumbol for technical work, the RNOH Musculoskeletal Pathology Biobank biobank team for consenting patients and accessing samples and the UCL CRUK Cancer Centre Pathology Core Facility for immunohistochemistry. We thank patients for participating in our research and the clinical teams involved in their care.

## Author contributions

M.W.F. and W.M. performed data analyses. M.D.Y. and P.S.T. contributed to data analysis. C.R.A., L.Z., A.M.F., R.T., F.A. and W.M. contributed to design of FISH experiments and supplied BAC clones. H.Y., F.B. and A.-C.S. performed FISH analyses. E.M. and F.B. performed immunostaining. A.V. performed the fusion validation with PCR and Sanger sequencing. J.D. contributed to allele-specific expression analysis. A.M.F., W.M., R.T. and F.A. curated and reviewed the samples, clinical data and/or provided clinical expertise. M.T., N.P., J.D. and A.E.G. contributed to discussions. M.W.F., S.B. and A.V. contributed to figure design. A.M.F., P.V.L., S.B., M.R.S. and P.J.C. directed the research. M.W.F, S.B. and A.M.F. wrote the manuscript.

## Additional information

**Competing interests:** The authors declare no competing interests.

