## [Peer Review File · Nature Communications]

Reviewers' comments:

Reviewer #1 (Remarks to the Author):

This interesting paper describes the first observation of rearrangements in the FOS and FOSB genes in a benign connective tissue tumors--osteoblastoma and osteoid osteoma. The data are convincing and confirmed using an independent method in a replication set of substantial size, with the mutations identified in the majority of cases. As the authors indicate, the frequency of rearrangement reaches the point where it could be used to supplement histology as a method to confirm diagnosis where required. As a general statement, clinical doubt is uncommon, owing to the distinctive pathologic and clinical presentation of these diseases.

The authors are not able to identify any 3' coding sequence arising as a consequence of the fusion. As a minor point, if the RNA sequence data were available, it would be good to provide this with the predicted codon usage to demonstrate the absence of meaningful 3' sequence. If this data were not available, this MS would be strengthened further by a technique such as capture seq that pulls down fusion transcripts using as bait the FOS exotic sequences to substantiate this observation.

Reviewer #2 (Remarks to the Author):

Ever since it was reported that the retroviral oncogene v-fos can induce bone tumours in mice, it has been a long-standing open question, whether alterations of FOS and subsequent dysregulation of c-Fos in mesenchymal cells can be causative for human bone cancers. This paper reports that FOS and FOSB rearrangements are found in human benign bone forming tumours, such as osteoblastoma and osteoid osteoma, which may be exploited as diagnostic markers.

The authors performed exome sequencing and found evidence for rearrangements in FOS and FOSB in 5 osteoblastoma and 1 osteoid osteoma. The authors propose that these rearrangements produce abnormal FOS transcripts with stop codons near breakpoints, and a fusion gene connecting PPP1R10 to FOSB. In addition, the authors show by FISH analyses with 55 patients that a high fraction of osteoblastoma display FOS/FOSB rearrangements and detected elevated nuclear reactivity of FOS by IHC. There are several points which need to be addressed, if this paper should be considered for publication:

1. Most importantly, there is no attempt to molecularly or functionally characterize the putatively expressed mutant Fos proteins. Human cDNA for truncated FOS mimicking at least 2-3 of the observed mutants should be generated and transfected into bone cells. These should be functionally analysed to verify that such truncated proteins are expressed and, in analogy to the v-fos gene (where the cDNA is of murine origin), more stable and/or more transcriptionally active. This will answer the question, if the resulting protein is indeed more stable and can this lead to some proliferative effects in vitro.
2. IHC for FOS in the positively stained samples is performed with an N-terminal specific antibody. Even if the somatic re-arrangements occurs in the two alleles of FOS, staining with a FOS N-terminal specific antibody covering the "missing region" should be shown, as it would be informative for FOS expression of the un-rearranged (wt) allele and/or demonstrate that the positive signal is indeed specific to the FOS C-terminal truncated protein.
3. IHC for FOS in the 3/6 FISH negative samples is mentioned on page 6, but the data should be shown, along with staining for FOSB in the FOS mutant samples (and vice versa a staining for FOS in the FosB sample).
4. Although the finding of a PPP1R10-FOSB fusion is new in this type of tumor and may serve as a candidate diagnostic factor for osteoblastoma and osteoid osteoma, the FOSB breakpoint is only detected in 1 case, and the expression of FOSB is negative by FOSB IHC (Supp. Fig. 2b). These data

are too preliminary with n=1 and should be substantiated with additional attempts to analyse FOSB expression or find additional cases. Alternatively, the authors should functionally characterize the resulting PPP1-FOSB fusion in vitro similar to point 1.

5. The very recent paper by Ijzendoorn et al. in JBC must be cited, since these authors functionally analyse the 1st mutant Fos protein identified in human vascular tumours, which is also a C-terminal truncation of FOS. The related papers cited in the study on FOS/FOSB in human tumours should be better discussed and in particular the interesting peculiarity of almost identical somatic rearrangement in FOS and FOSB occurring in two types of cancers. For example, since some of the previously observed FOS rearrangements were observed in epithelioid hemangioma of the bone and found more prevalent in intraosseous lesions, could it be that some of these tumours share a similar cell of origin? Minor point: The authors should clearly indicate that the genetic alterations found in FOS genes are somatic and mono-allelic.

Reviewer #3 (Remarks to the Author):

The authors employed WGS and RNA-Seq to discover recurrent FOS/FOSB rearrangements in six osteoblastoma patients, and further proved that FOS/FOSB rearrangements are ubiquitous and specific to osteoblastoma, likely can be used as the biomarker of this benign tumor. The result is novel and potentially translational. I have several comments.

*** Major:

1) The authors should provide citations for all tools used in their analysis (Caveman, Pindel, Brass, Ascat, HTSeq, Tophat, etc.).

2) The authors should provide the details of the WGS and RNA-seq sequencing QC statistics in a summary table.

3) The authors did not provide enough details on the SNVs, INDELS, and SVs identified from their patients. Read count, depth, variant allele frequency, amino acid change are all helpful for the readers to interpret the results and perform follow up studies.

4) Particularly, the authors should provide validation results for FOS/FOSB rearrangements, ideally by targeted sequencing, but this also depends on how much depth and the number supporting reads in WGS and if they could distinguish true somatic calls from false positives or germline variants. Why were there only two SVs involving FOS in Table S5? Is this table incomplete?

5) I am not sure how the HTSeq results were used to assess the expression of the genes involved in FOS/FOSB rearrangement. I also wonder what gene annotation was used for generating read counts and how the fusion transcript count was generated? HTSeq is quite conservative and may discard many ambiguous reads that disagree with the gene annotation.

6) The authors only provided RNA-Seq coverage data for the fusion partners of FOS. To better understand the transcriptional consequence of the fusion, the authors should also include the RNA-Seq coverage map for both FOS and its fusion partner genes/regions in individual tumors. Is there any allelic expression imbalance and how does the expression of the fusion transcript compared with control normal tissues? This would potentially reveal that the FOS rearrangements indeed resulted in overexpression as a mechanism of FOS rearrangement indicated by the authors.

7) Similarly, for the FOSB fusion, the authors should include RNA-Seq coverage map for the fusion transcript to further elaborate their claim at line 80 "The expression of the FOSB fusion gene is

brought under the control of the PPP1R10 promoter”.

*** Minor:

Since loss of chr22 was not observed, I wonder whether this may be due to low tumor purity or marginal signal? Thus I recommend to include all data points used to infer copy number (eg. log2ratio of read depths between tumor and normal or similar metrics) in the copy number profiles presented in Fig S1.

Line 109: “In these 55 samples, we found FOSB and FOS breakapart signals in 1 and 48 (87%) tumours respectively (Figures 1c).”. I don’t think Fig 1c has the data to support this result.

Table header is repeated in Table S2 (row 316 of the Excel sheet).

Table S2 and S3 are overlapped, the authors should consider merging them.

Line 10-74: In “... However, no coding sequence from the FOS fusion partner was transcribed”, did the authors mean “translated” instead of “transcribed”?

Line 94: “active promotor”. I suggest to change “promotor” to “promoter”.

Response to Reviewers

We would like to thank the Reviewers for taking the time to read our report. We are most grateful for the comments which have helped us to improve considerably the manuscript. Our point by point response is outlined below.

Reviewer #1

#	Reviewer's comment	Our response
1.1	This interesting paper describes the first observation of rearrangements in the FOS and FOSB genes in a benign connective tissue tumors--osteoblastoma and osteoid osteoma. The data are convincing and confirmed using an independent method in a replication set of substantial size, with the mutations identified in the majority of cases. As the authors indicate, the frequency of rearrangement reaches the point where it could be used to supplement histology as a method to confirm diagnosis where required. As a general statement, clinical doubt is uncommon, owing to the distinctive pathologic and clinical presentation of these diseases.	We thank the reviewer for their enthusiasm for our study.
1.2	The authors are not able to identify any 3' coding sequence arising as a consequence of the fusion. As a minor point, if the RNA sequence data were available, it would be good to provide this with the predicted codon usage to demonstrate the absence of meaningful 3' sequence. If this data were not available, this MS would be strengthened further by a technique such as capture seq that pulls down fusion transcripts using as bait the FOS exotic sequences to substantiate this observation.	We apologise for the lack of clarity. We do have RNA sequences, from which we had predicted stop codon insertion. As suggested by the Reviewer, we now include the predicted peptide sequence of each fusion (see Supplementary Figure 2). Each fusion had been predicted from DNA sequences. We have now validated these fusions using Sanger sequencing on RNA (i.e. cDNA; see Supplementary Figure 2).

Reviewer #2

#	Reviewer's comment	Our response
2.1	Ever since it was reported that the retroviral oncogene v-fos can induce bone tumours in mice, it has been a long-standing open question, whether alterations of FOS and subsequent dysregulation of c-Fos in mesenchymal cells can be causative for human bone cancers. This paper reports that FOS and FOSB rearrangements are found in human benign bone forming tumours, such as osteblastoma and osteoid osteoma, which may be exploited as diagnostic markers. The authors performed exome sequencing and found evidence for rearrangements in FOS and FOSB in 5 osteblastoma and 1 osteoid osteoma. The authors propose that these rearrangements produce abnormal FOS transcripts with stop codons near breakpoints, and a fusion gene connecting PPP1R10 to FOSB. In addition, the authors show by FISH analyses with 55 patients that a high fraction of osteblastoma display FOS/FOSB rearrangements and detected elevated nuclear reactivity of FOS by IHC.	We thank the reviewer for their time in reviewing our study and providing valuable comments which we have addressed as outlined below.
2.2	Most importantly, there is no attempt to molecularly or functionally characterize the putatively expressed mutant Fos proteins. Human cDNA for truncated FOS mimicking at least 2-3 of the observed mutants should be generated and transfected into bone cells. These should be functionally analysed to verify that such truncated proteins are expressed and, in analogy to the v-fos gene (where the cDNA is of murine origin), more stable and/or more transcriptionally active. This will answer the question, if the resulting protein is indeed more stable and can this lead to some proliferative effects in vitro.	We certainly agree with the reviewer that our study does not provide a functional evaluation of FOS aberrations in osteblastoma, although some hypotheses are put forward. Such experiments would require a dedicated, prolonged effort that is beyond the scope of our study. Our work is an integrative genomic study of osteblastoma that provides a definitive genomic aberration underpinning osteblastoma.

2.3	IHC for FOS in the positively stained samples is performed with an N-terminal specific antibody. Even if the somatic re-arrangements occur in the two alleles of FOS, staining with a FOS N-terminal specific antibody covering the “missing region” should be shown, as it would be informative for FOS expression of the un-rearranged (wt) allele and/or demonstrate that the positive signal is indeed specific to the FOS C-terminal truncated protein.	We thank the reviewer for giving us the opportunity to elaborate on the IHC finding and for proposing this experiment. We do not find evidence of biallelic mutation of FOS. Only one rearrangement per sample was found. The genomic pattern of mutation in FOS indicates that FOS operates as a dominant cancer gene, i.e. an oncogene. This interpretation is corroborated by the functional studies of the mutation performed in vascular tumours which the Reviewer had kindly highlighted to us (see point 2.6). As is the case in virtually all oncogenes, whilst there can be a bias towards expression of the mutated allele, one would generally expect expression of the wildtype allele also which invariably will lead to some wild type protein being present (even if the mutant one were more stable). Bearing this in mind, we performed the IHC experiment the reviewer suggested and provide these for the Reviewer as Reviewer Figure 1. We compared N- and C-terminal [Santa Cruz-8047] staining side by side in the 5 FOS mutant tumours of the discovery cohort, with appropriate positive and negative control tissue (colonic epithelium and FOSB mutant cases, respectively). Interestingly, we found no C-terminal immunoreactivity in 3/5 FOS-mutant osteoblastoma cases which exhibited intense N-terminal staining. This would suggest a strong bias for expression of the mutant protein. In 2/5 cases we found both N- and C-terminal staining. As the epitope used to raise the C-terminal antibody (FOS aa 210–335) spans the breakpoints, it is likely that in these two cases, at least part of the epitope remains. We can not exclude expression of the wild-type FOS allele in these two cases. An alternative interpretation could be that in the two positive cases the antibody recognizes conserved parts of the homologous FRA1, FRA2 or FOSB proteins. These proteins are 25-40% identical to FOS across its most C-terminal part. Considering this degree of homology any antibody designed for reactivity against the truncated portion of FOS will be non-specific. Based on this, we feel uncomfortable drawing strong conclusions from relative N- vs C-terminal staining. Further, whether or not the wild type allele is expressed does not affect the main conclusions of this paper, namely, that:  (i) FOS is mutated in nearly all osteoblastomas, as demonstrated by genomic techniques. (ii) FOS IHC is exceptionally intense in osteoblastoma, compared to control tissues and compared to osteosarcoma, even when the antibody is applied at 1/10th of the normal concentration (see Supplementary Figure 4a).
-------------------	---	--

2.4	IHC for FOS in the 3/6 FISH negative samples is mentioned on page 6, but the data should be shown, along with staining for FOSB in the FOS mutant samples (and vice versa a staining for FOS in the FosB sample).	We are grateful to the reviewer for raising this oversight. We now include in the manuscript, as follows: IHC for FOS in the 3/6 negative cases > added to Suppl. Figure 4b FOS immunostaining in the FOSB mutant samples > added to Suppl. Figure 4c FOSB immunostaining in the FOS mutant cases > provided as Reviewer Figure 2
2.5	Although the finding of a PPP1R10-FOSB fusion is new in this type of tumor and may serve as a candidate diagnostic factor for osteblastoma and osteoid osteoma, the FOSB breakpoint is only detected in 1 case, and the expression of FOSB is negative by FOSB IHC (Supp. Fig. 2b). These data are too preliminary with n=1 and should be substantiated with additional attempts to analyse FOSB expression or find additional cases. Alternatively, the authors should functionally characterize the resulting PPP1-FOSB fusion in vitro similar to point 1.	We thank the reviewer for giving us the opportunity to elaborate on FOSB mutation in osteblastoma. Clearly, this is a rare finding. As the reviewer says, we identified a single case in the discovery cohort. However, FISH results from the extension cohort validate this finding by revealing a second case (case 50; see Supplementary Table 1). Our extended group of cases is substantial for a rare disease (see also comment 1.1 of Reviewer 1), and therefore acquiring more cases would be extremely difficult. If the incidence of FOSB fusions was 2/61 (3.3%) a further case series of at least 150 would be needed. We estimate that the annual osteblastoma resection rate in the UK is only ~25 and most of these would not be suitable for genomic analysis because of inappropriate decalcification. It is therefore likely to take a further 10-15 years to accumulate the required samples. Similar FOSB fusion events are seen at a higher frequency in vascular tumours. The functional impact of these FOSB fusion events has not been fully explored in these cases. Nevertheless, our conclusions regarding promoter hijacking are in keeping with published reports, in particular the cited studies of Walther et al and of van Ijzendoorn et al.
2.6	The very recent paper by Ijzendoorn et al. in JBC must be cited, since these authors functionally analyse the 1st mutant Fos protein identified in human vascular tumours, which is also a C-terminal truncation of FOS. The related papers cited in the study on FOS/FOSB in human tumours should be better discussed and in particular the interesting peculiarity of almost identical somatic rearrangement in FOS and FOSB occurring in two types of cancers. For example, since some of the previously observed FOS rearrangements were observed in epithelioid hemangioma of the bone and found more prevalent in intraosseous lesions, could it be that some of these tumours share a similar cell of origin?	We are very grateful for these valuable insights raised by the reviewer. We are glad to incorporate reference to this recent publication by van Ijzendoorn et al, which was published just as we had submitted our manuscript. The second paragraph of our discussion now highlights these findings, see lines 152-155 of the manuscript: “Furthermore, ubiquitin-independent proteasomal degradation rapidly depletes the wild-type FOS protein. The C-terminal truncations seen in epithelioid haemangioma have recently been shown to protect FOS from degradation.” This provides excellent support for our supposition that the genomic events we had summarised in Figure 1d (which includes van Ijzendoorn’s case) lead to increased FOS protein levels. We now also labelled the C-terminal proteasomal degradation domain in Figure 1d. Further functional characterisation, beyond proteasomal degradation and effects on endothelial sprouting, are clearly required. These include exploration of transcript stability and the

		possibility of altered DNA binding and transcription factor activity. This, we consider to be beyond the scope of this study. We agree that speculation about common cells of origin is intriguing, though we are cautious about overstating this without substantial evidence. We believe that the penultimate sentence of our manuscript hints at this possibility but acknowledges the need for further work: “Our findings also draw an intriguing parallel between bone-forming tumours and a subset of vascular tumours indicating possible shared developmental pathways.”
2.7	Minor point: The authors should clearly indicate that the genetic alterations found in FOS genes are somatic and mono-allelic.	We thank you for this clarification and we now state this explicitly in the results in our manuscript, see line 80-83: “A single FOS or FOSB breakpoint was confirmed in each sample, suggesting that these were mono-allelic rearrangements. There was no evidence of similar rearrangements in paired normal tissue samples, confirming that they were somatic.”

Reviewer #3

#	Reviewer's comment	Our response
3.1	The authors employed WGS and RNA-Seq to discover recurrent FOS/FOSB rearrangements in six osteoblastoma patients, and further proved that FOS/FOSB rearrangements are ubiquitous and specific to osteoblastoma, likely can be used as the biomarker of this benign tumor. The result is novel and potentially translational.	We are very grateful for the reviewer's helpful remarks which have allowed us to improve the manuscript.
3.2	The authors should provide citations for all tools used in their analysis (Caveman, Pindel, Brass, Ascat, HTSeq, Tophat, etc.).	We are happy to have included references or github links, in the revised manuscript.
3.3	The authors should provide the details of the WGS and RNA-seq sequencing QC statistics in a summary table.	We have included all QC statistics in a new Supplementary Table 9 .
3.4	The authors did not provide enough details on the SNVs, INDELS, and SVs identified from their patients. Read count, depth, variant allele frequency, amino acid change are all helpful for the readers to interpret the results and perform follow up studies.	We have expanded on Supplementary Tables 2,3 and 5 to include these data.
3.5	Particularly, the authors should provide validation results for FOS/FOSB rearrangements, ideally by	We apologise for the lack of clarity.

	targeted sequencing, but this also depends on how much depth and the number supporting reads in WGS and if they could distinguish true somatic calls from false positives or germline variants. Why were there only two SVs involving FOS in Table S5? Is this table incomplete?	Fusions were identified from DNA sequences and validated by RNA sequencing. Please see the new Supplementary Fig. 2. This now includes validation by Sanger sequencing of cDNA for FOS mutant cases. Regarding Supplementary Table 5, this is a “purist’s” table of rearrangements. I.e., we only included rearrangements here that passed stringent filtering. It is intended as a formal catalogue of rearrangements for any downstream analyses someone may want to perform. The missing FOS rearrangements were discovered in DNA reads by “digging” into sequencing data deeply. I.e. we manually extracted discordantly mapped reads from DNA sequences and then validated these raw calls by RNA sequencing. For your scrutiny, we included all of these reads in Supplementary Tables 6 (DNA discordant reads), 7 (DNA split reads spanning the breakpoint junction), and 8 (RNA i.e. cDNA reads spanning the junction).
3.6	I am not sure how the HTSeq results were used to assess the expression of the genes involved in FOS/FOSB rearrangement. I also wonder what gene annotation was used for generating read counts and how the fusion transcript count was generated? HTSeq is quite conservative and may discard many ambiguous reads that disagree with the gene annotation.	We apologise if the original methods section misrepresented this normalisation step. We have amended line 225-227 to clarify this: “The per-base coverage in these regions therefore reveals a clear peak, present only in that tumour sample, demonstrating expression of aberrant transcripts (normalised by the mean of HTSeq counts $\times 10^3$).” HTSeq counts were only used to generate a proxy of useful library size in order to normalise RNAseq coverage plots shown in Figure 1. Coverage data was generated using bedtools coverage (split) for per base coverage statistics. Per base coverage was then normalised by dividing by the mean HTSeq count in the sample to account for library size. Conservative counting will not have an impact as these counts are merely used as normalising factors between the different samples. We used the default settings for HTseq count, counting the union of exon coordinates for each genomic feature. We used an ensembl features file (gtf) which were appropriate for RNAseq alignment as follows: Genome-build GRCh37.p13 Genome-version GRCh37 Genome-date 2009-02 Genome-build-accession NCBI:GCA_000001405.14 Genebuild-last-updated 2013-09 Ensembl release 75
3.7	The authors only provided RNA-Seq coverage data for the fusion partners of FOS. To better understand the transcriptional consequence of the fusion, the authors should also include the RNA-Seq coverage map for both FOS and its fusion partner genes/regions in individual tumors. Is there any allelic expression imbalance and how does the	We are very grateful to the reviewer for these helpful remarks. We now include RNAseq coverage plots for FOS in the new Supplementary Figure 3a. Please note that the purity of our samples is relatively low and, as a result, changes in (allelic) gene expression in the tumour cells may be difficult to detect. With regards allelic expression imbalance, 2/5 samples harbouring no heterozygous SNPs in

	expression of the fusion transcript compared with control normal tissues? This would potentially reveal that the FOS rearrangements indeed resulted in overexpression as a mechanism of FOS rearrangement indicated by the authors.	FOS. In the three samples that do have heterozygous SNPs (see Supplementary Figure 3), we do not find clear evidence of allelic imbalance. This might be due to lack of sensitivity as a result of poor cellularity or, it may indicate that FOS mutation has a predominant effect at the protein level. We have amended the second paragraph of the discussion in the manuscript to expand on these implications, particularly in the light of the recent publication from van Ijzendoorn et al. Specifically we include the following two referenced sentences at lines 152-155: “Furthermore, ubiquitin-independent proteasomal degradation rapidly depletes the wild-type FOS protein. The C-terminal truncations seen in epithelioid haemangioma have recently been shown to protect FOS from degradation.”
3.8	Similarly, for the FOSB fusion, the authors should include RNA-Seq coverage map for the fusion transcript to further elaborate their claim at line 80 “The expression of the FOSB fusion gene is brought under the control of the PPP1R10 promoter”.	See 3.7 above and new Supplementary Figure 3b. There is a subtle increase in RNAseq coverage after the breakpoint for the FOSB fusion sample (PD7525). This again is compromised by the low tumour purity in all these samples. In addition, there appears to be allelic imbalance in intron 1 alone. This intron-specific imbalance may imply a higher level of transcriptional activity¹. 1. Alkallas, R., Fish, L., Goodarzi, H. & Najafabadi, H.S. Inference of RNA decay rate from transcriptional profiling highlights the regulatory programs of Alzheimer's disease. Nat Commun 8, 909 (2017).
3.9	Since loss of chr22 was not observed, I wonder whether this may be due to low tumor purity or marginal signal? Thus I recommend to include all data points used to infer copy number (eg. log2ratio of read depths between tumor and normal or similar metrics) in the copy number profiles presented in Fig S1.	We are very grateful to the reviewer for this helpful suggestion. We now show the data in Supplementary Figure 1 . It is correct to assert that low tumour purity has hampered finding copy number aberrations. We have scrutinised in detail the relevant regions of chromosome 22, which were previously reported to carry homozygous deletions. We do not find any evidence of somatic deletions in the region previously highlighted (~15-40MB). There are only possible regions of germline copy number gain.
3.10	Line 109: “In these 55 samples, we found FOSB and FOS breakapart signals in 1 and 48 (87%) tumours respectively (Figures 1c).”. I don't think Fig 1c has the data to support this result.	We are grateful for pointing out this typographical error and this of course, should state Supplementary Table 1. We have amended the manuscript at line 112-114: “In these 55 samples, we found FOSB and FOS breakapart signals in 1 and 48 tumours, respectively (89% in total; Supplementary Table 1).”
3.11	Table header is repeated in Table S2 (row 316 of the Excel sheet).	This has been corrected.
3.12	Table S2 and S3 are overlapped, the authors should consider merging them.	This is intentional, in order to make it easier for the reader to scrutinise the more relevant coding variants, reported in Supplementary Table 3 . We have altered the table headers to make this clearer.

3.13	Line 10-74: In "... However, no coding sequence from the FOS fusion partner was transcribed", did the authors mean "translated" instead of "transcribed"?	We are grateful for this being highlighted. The wording was clearly misleading and has been amended at line 90-91 of the manuscript: "However, these aberrantly transcribed sequences did not include any known exonic sequence."
3.14	Line 94: "active promotor". I suggest to change "promotor" to "promoter".	Again, we thank you for spotting this typographical error.

REVIEWERS' COMMENTS:

Reviewer #1 (Remarks to the Author):

Good job of responding--well done.

Reviewer #2 (Remarks to the Author):

I have now carefully looked at the revised manuscript. Overall, the manuscript has slightly improved, however the main points I raised in my review were not addressed! To provide 2 examples:

1) No functional data are provided for some of the Fos mutant proteins and the suggestions for a quick in vitro experiment were ignored.

2) The FosB breakpoint mutant analyses: $n=1/2$; if I understand the data correctly, FosB expression is negative in the patient sample! Why say the 'functional impact has not been fully explored' and call the sample FosB positive, when by IHC in Suppl Figure the staining is negative etc.

Overall, I can't see that this study is of sufficient depth and significance or conceptual advance to be a strong candidate for being considered in Nature Comm.

Reviewer #3 (Remarks to the Author):

The authors have addressed all my concerns. Thanks.

Recurrent rearrangements of FOS and FOSB define osteblastoma – NCOMMS-17-30837-A

Response to Reviewers

We would like to thank the Reviewers for taking further time to read our responses. We are most grateful for the further comments to which our response is outlined below.

Reviewer #1

#	Reviewer's comment	Our response
1.1	Good job of responding--well done.	We thank the reviewer again for their enthusiasm for our study.

Reviewer #2

#	Reviewer's comment	Our response
2.1	No functional data are provided for some of the Fos mutant proteins and the suggestions for a quick in vitro experiment were ignored.	We apologise that the reviewer considered his comments ignored. To reiterate our previous response to this particular point on functional evaluation, we agree with the reviewer that our study does not provide a functional evaluation of FOS aberrations in osteblastoma, although some hypotheses are put forward. Such experiments would require a dedicated, prolonged effort that is beyond the scope of our study. Our work is an integrative genomic study of osteblastoma that provides a definitive genomic aberration underpinning osteblastoma.
2.2	The FosB breakpoint mutant analyses: n=1/2; if I understand the data correctly, FosB expression is negative in the patient sample! Why say the 'functional impact has not been fully explored' and call the sample FosB positive, when by IHC in Suppl Figure the staining is negative etc.	We thank the reviewer for raising this point regarding the FOSB IHC. To clarify 1/6 of our discovery cohort had incontrovertible FOSB fusion, demonstrated by three independent genomic lines of evidence: DNA reads split by the fusion, DNA reads spanning the fusion, and cDNA reads also spanning the fusion. All of these reads are supported in Supplementary Tables 6,7 and 8. A further case in our validation cohort (1/55) had evidence of a DNA breakapart in FOSB by FISH. Therefore 2 cases in total contained a FOSB breakapart. FOSB immunohistochemistry was not found to be reliable in osteblastoma, as is demonstrated in Supp Figure 4c. This is consistent with the most extensive evaluation of this antibody. Hung et al found, in Pseudomyogenic haemangi endothelioma, which has a much higher prevalence of FOSB rearrangement, that their two decalcified cases were both

		negative for FOSB immunoreactivity. As osteoblastoma require routine decalcification we considered that this assay would be uninformative for this disease. We have clarified this in the manuscript by an additional sentence at line 124-125: “FOSB immunohistochemistry was uninformative in osteoblastoma, consistent with previous experience with decalcified tumours (Supplementary Fig. 4c)⁹.” In addition, we have amended the legend and headings of Supplementary Figure 4 to emphasise that these are positive control FOS/FOSB fusion positive cases, established with strong genomic evidence.
--	--	---

Reviewer #3

#	Reviewer's comment	Our response
3.1	The authors have addressed all my concerns. Thanks.	We are again grateful for reviewer 3's contributions to improving our submission.